# Hydrologic Modeling for Sustainable Water Resources Management in Urbanized Karst Areas

**DOI:** 10.3390/ijerph16142542

**Published:** 2019-07-16

**Authors:** Hugo Henrique Cardoso de Salis, Adriana Monteiro da Costa, João Herbert Moreira Vianna, Marysol Azeneth Schuler, Annika Künne, Luís Filipe Sanches Fernandes, Fernando António Leal Pacheco

**Affiliations:** 1Departamento de Geografia, Universidade Federal de Minas Gerais, Av. Antônio Carlos, 6.627 Pampulha, Belo Horizonte 31270-901, Minas Gerais, Brazil; 2EMBRAPA Milho e Sorgo, Rodovia MG-424, Km 45 Caixa Postal: 285 ou 151, Sete Lagoas 35701-970, Minas Gerais, Brazil; 3EMBRAPA Solos, Rua Jardim Botânico, nº 1.024, Bairro Jardim Botânico, Rio de Janeiro 22460-000, Brazil; 4Geographic Information Science Group, Institute of Geography, Friedrich Schiller University Jena, 07749 Jena, Germany; 5Centro de Investigação e Tecnologias Agroambientais e Biológicas, Universidade de Trás-os-Montes e Alto Douro, Ap 1013, 5001-801 Vila Real, Portugal; 6Centro de Química de Vila Real, Universidade de Trás-os-Montes e Alto Douro, Ap 1013, 5001-801 Vila Real, Portugal

**Keywords:** karst aquifers, recharge, land use and occupation, waterproofing, hydrologic modeling, JAMS J2000, water resources management, sustainability

## Abstract

The potential of karst aquifers as a drinking water resource is substantial because of their large storage capacity gained in the course of carbonate dissolution. Carbonate dissolution and consequent development of preferential paths are also the reasons for the complex behavior of these aquifers as regards surface and underground flow. Hydrological modeling is therefore of paramount importance for an adequate assessment of flow components in catchments shaped on karsts. The cross tabulation of such components with geology, soils, and land use data in Geographic Information Systems helps decision makers to set up sustainable groundwater abstractions and allocate areas for storage of quality surface water, in the context of conjunctive water resources management. In the present study, a hydrologic modeling using the JAMS J2000 software was conducted in a karst area of Jequitiba River basin located near the Sete Lagoas town in the state of Minas Gerais, Brazil. The results revealed a very high surface water component explained by urbanization of Sete Lagoas, which hampers the recharge of 7.9 hm^3^ yr^−1^ of storm water. They also exposed a very large negative difference (−8.3 hm^3^ yr^−1^) between groundwater availability (6.3 hm^3^ yr^−1^) and current groundwater abstraction from the karst aquifer (14.6 hm^3^ yr^−1^), which is in keeping with previously reported water table declines around drilled wells that can reach 48 m in old wells used for public water supply. Artificial recharge of excess surface flow is not recommended within the urban areas, given the high risk of groundwater contamination with metals and hydrocarbons potentially transported in storm water, as well as development of suffosional sinkholes as a consequence of concentrated storm flow. The surface component could however be stored in small dams in forested areas from the catchment headwaters and diverted to the urban area to complement the drinking water supply. The percolation in soil was estimated to be high in areas used for agriculture and pastures. The implementation of correct fertilizing, management, and irrigation practices are considered crucial to attenuate potential contamination of groundwater and suffosional sinkhole development in these areas.

## 1. Introduction

It is estimated that 3% to 6% of the Brazilian territory is covered by carbonate rocks and that in the state of Minas Gerais this percentage is around 22%. These areas are likely to be exploited for drinking water because their groundwater resources are substantial, proportionate to karst development that expands the aquifer storage capacity [1,2]. The evaluation of groundwater resources is complex in general, because water balance parameters, flow equations, and other control variables of abundance and movement of water in catchments and aquifers are spatially heterogeneous. The task becomes particularly difficult in karst areas, because developing karsts becomes progressively more heterogeneous as regards water movement on the surface and subsurface. The development of preferential flow paths in the course of carbonate dissolution (sinkholes, dissolution conduits) is the paramount example of such complexity. A common way to accomplish groundwater resource evaluation is through hydrologic modeling at the catchment scale. The number of studies specifically related to the hydrologic modeling of karst systems is relatively scarce [3,4,5,6], despite the great concern of many researchers as regards the risk of groundwater contamination in these aquifers [7,8]. This is also motivation to explore the hydrologic behavior of these aquifers.

Hydrological models are tools that perform a mathematical representation of hydrological processes, such as infiltration of water into the soil, recharge of aquifers, runoff and drainage network flow [9,10], as well as hydrochemical processes such as weathering or contaminant transport [11,12,13,14,15,16,17,18,19]. They can be used to supply and/or supplement information from a particular location as close as possible to the actual hydrological dynamics of a river basin. They can also help watershed managers in the control of extreme events such as floods [20,21,22,23,24,25], or in the assessment of water resource availability [26,27,28,29] and threats to water quality [30,31,32,33,34,35,36,37,38,39,40,41,42]. Among numerous examples, one can refer the following models: TBHM—Topography Based Hydrological Model [43], HES—European Hydrological System [44], SWAT—Soil and Water Assessment Tool [45], THMB—Terrestrial Hydrologic Model with Biogeochemistry [46]. In addition to these models, the JAMS J2000 framework (http://jams.uni-jena.de/) can be highlighted, as an open-source, process-oriented hydrological model that presents climatic data regionalization, volumetric soil water balance, volumetric groundwater balance, and propagation of flows in water courses. Several authors have used this model to simulate different types of environments, highlighting its adaptability to complex geological settings including karsts [47,48].

Despite its importance, the application of models such as JAMS J2000 to explore the hydrologic behavior of karst basins is still uncommon in the territory of Minas Gerais. The information derived from this modeling, such as groundwater resources and degree of aquifer vulnerability, would support planning initiatives to prevent overexploitation and possible impacts from anthropogenic activities on the aquifer. Moreover, distributed models are prepared to delineate the spatial distribution of hydrological processes, quickly and inexpensively, allowing assessments of groundwater resources and aquifer vulnerability at local and regional scales. Thus, the objective of this work was to perform the calibration and validation of a hydrological model in a karst region of Minas Gerais (the Jequitiba River basin), using the JAMS J2000 framework, and interpret the results from a water resources management perspective. The Jequitiba basin was selected because the largest town located inside the basin (Sete Lagoas) has explored for decades since the 1980s a karst aquifer with evident signs of overexploitation (e.g., suffosional sinkholes).

## 2. Materials and Methods

### 2.1. Study Area

The study area is a portion of the Jequitibá River Basin with approximately 24,000 ha, located in the central region of Minas Gerais state, which comprises portions of various municipalities, namely Capim Branco, Prudente de Morais, and Sete Lagoas. The latter is the largest town, with 145,729 inhabitants that represent 97.6% of the entire catchment population. The basin is located between the geographic coordinates X = 573,198 to 594,872 m and Y = 7,859,607 to 7,836,875 m, referring to the SIRGAS 2000 geodetic datum and UTM 23 South projection, while the altitudes range from 629 to 932 m (Figure 1a,b). According to Koppen’s classification, the climate is subtropical (Cwa), characterized by dry winters and hot summers. The mean annual rainfall in the 2000–2018 period was 1291.2 mm, while the mean temperatures varied from 18 ºC in July and 24 ºC in January–February, with a mean annual value of 21.8 ºC. The geology is characterized by a predominance of carbonate rocks (Figure 1c). The stratigraphic sequence comprises orthogneisses, granites and migmatites that represent an Archean crystalline basement. These igneous and metamorphic rocks were overlaid by Neoproterozoic carbonate rocks of the Bambuí Group, namely calcite and dolomite limestones from the Sete Lagoas Formation, and pelitic rocks with interlayered carbonates from the Serra de Santa Helena Formation. The igneous, metamorphic and sedimentary rocks were later covered by terrigenous rocks composed of alluvium, colluvium, and terrace sediments along and lateral to the main water courses [49]. Figure 1c exposes the predominance of limestones (42.7%), followed by pelitic rocks interlayered with carbonates (28.7%), Archean basement (24.3%), and colluvium (4.4%), in the studied part of the Jequitiba River basin [50]. The comparison of Figure 1c with Figure 1d (soil map [51]) suggests the following genetic associations between litho types and soil types: the Archean rocks as well as pelitic rocks cropping out in the catchment lowlands have weathered to cambisols (36.7% coverage within the basin); pelitic rocks cropping out in the catchment highlands have weathered to neosols (12.2%); limestones and terrigenous rocks have weathered to latosols (49.6%). Land in the Jequitiba River basin is mostly used for anthropogenic activities, such as livestock pasturing or agriculture, which are distributed along the drainage network (Figure 1e). Forests occupy 15.8% of the area and urban areas 14.4% [52].

The Archean rocks developed fractured aquifers while the pelitic rocks interlayered with carbonates from the Serra de Santa Helena formation and the limestones from the Sete Lagoas formation developed fractured-karst aquifers and karst aquifers, respectively. Finally, the terrigenous rocks and the soil layer, which can be thick, developed porous aquifers [53]. Specific flows in these aquifers range from very low in the fractured aquifers (average: 0.52 m^3^ ha^−1^ m^−1^), low in the fracture-karst aquifers (average: 20.84 m^3^ ha^−1^ m^−1^), and high in the karst aquifers (can reach 264 m^3^ ha^−1^ m^−1^) (http://www.cprm.gov.br). Ground water from the karst aquifer is used for public water supply to the towns. A special mention is due to Sete Lagoas town because it represents 97.6% of all the population living in the basin. According to Pessoa [54], in Sete Lagoas the use of groundwater as the main resource for the public water supply started in the 1950s. In those days, Sete Lagoas town hosted approximately 25 000 inhabitants and groundwater was extracted from private and public cisterns installed in the saturated water table aquifer that extends from 10 to 50 m depth. As the population grew (in 1960, the inhabitants of Sete Lagoas were already larger than 40,000), these cisterns could no longer satisfy the public water demand, and consequently were progressively replaced by drilled wells installed in the karst aquifer up to 160 m, the maximum aquifer depth.

From the 1990s onwards, significant amounts of groundwater were withdrawn from the karst aquifer, because the population of Sete Lagoas as well as the water demand had grown considerably [55]. According to Pessoa [54], in 1993 the water for nearly 150,000 people consuming 200 L habitant^−1^ day^−1^ was supplied by 65 drilled wells with an average yield of 8.0 L s^−1^ (520 L s^−1^ of total yield). The pressure over the drilled wells was evaluated with 16 h of pumping every day and considered preoccupying. Besides, the quality of these resources was threatened because a domestic sewage system was lacking in the town. The situation of Sete Lagoas was re-evaluated in 2008 by Botelho [56], with similar conclusions. Twenty-five years after the evaluation of Pessoa, the number of drilled wells was raised from 65 to 94 (44% increase), keeping a similar average yield (7.8 L s^−1^), while the population of Sete Lagoas increased from 150,000 to 220,000 (47% increase). In 2014, Galvão et al. [57] evaluated the effects of pumping on the geometry of hydraulic heads within the area of Sete Lagoas where the number of drilled wells and pumping rates are larger. A hydraulically depressed area was delineated around the older wells (1942) where depths to the water ranged from 14 m post drilling to 62 in 2012 (48 m drawdown in 70 years). According to age versus drawdown data available in the study of Galvão, it is possible to estimate an average of 0.9 m yr^−1^ of drawdown within the depressed area, caused by excessive pumping. The study of Galvão also suggested the link of this hydraulic head depression to the development of suffosional sinkholes.

### 2.2. Databases and Software

The materials used in this study are indicated in Table 1 and comprised: (a) a Digital Elevation Model (DEM) ALOS PALSAR with a spatial resolution of 12.5 m [58]; (b) Sentinel-2 satellite images with a spatial resolution of 10 m [59]; (c) the soil map of Minas Gerais state at scale of 1:650,000 and corresponding data on hydraulic conductivity obtained from texture data per soil type [51]; (d) the geological map of Minas Gerais state at scale 1:1,000,000 [50]; (e) Climatic data from weather stations in the municipalities of Belo Horizonte (BH), Sete Lagoas (SL), Conceição do Mato Dentro (CMD), and Florestal (FLT) [60]; (f) Hydrometric data of station 41410000 [61]; (g) data from the Rural Environmental Registry (CAR) about administrative issues [62]; (h) population data relative to the studied area [55].

The software Hydrus 1D (https://www.pc-progress.com) was used to estimate hydraulic conductivity of soils based on soil texture data per soil type released with the 1/650,000 soil map. The SPRING software, version 4.3.5 (http://www.dpi.inpe.br/spring/english/), was used to interpret the satellite images and delineate land uses and occupations based on the regions approach. The JAMS J2000 framework (http://jams.uni-jena.de) was used to perform the hydrologic modeling, including the calibration and validation procedures. The Quantum GIS (https://www.qgis.org) was used to produce the thematic maps (e.g., Figure 1).

### 2.3. Hydrological Modeling

The hydrologic modeling was developed in six main steps (Figure 2): (1) pre-processing of climatic and stream flow data. The data records comprised the 2003–2016 period; (2) mapping of land use and occupation; (3) design of homogeneous hydrologic response units in the area; (4) parameterization of input data; (5) hydrologic modeling based on the JAMS J2000 framework within the calibration (2003–2011), validation (2012–2016) and whole data (2003–2016) periods. The whole data period (14 years) was defined on the basis of available data. Within this period, a larger time span (9 years) was ascribed to the calibration period to allow improved hydrologic parameters, while the time window for validation was restricted to the remaining 5 years; (6) performance analysis of calibrated models based on comparisons between observed and simulated hydrographs as well as in goodness-of-fit indices.

In the first stage, the variables precipitation, temperature, relative humidity, hours of sunshine, wind speed, as well as daily stream flow data, were organized in a series of Excel spreadsheets. In turn, these spreadsheets were submitted to the online platform INTECRAL RBIS for conversion into files in the specific format of JAMS J2000.

In the second stage, the Sentinel-2 satellite images were clipped to extract the study area, and verified for radiometric, geometric, and geographical consistency. The clipped images were then interpreted and classified within regions to produce a land use and occupation map for the study area. The classification comprised classes for cultivated area, urban area, native vegetation (Cerrado biome), water bodies, planted forest, managed pasture, exposed soil, and herbaceous vegetation.

The homogeneous hydrologic response units (HRU) were delineated in the third step, using the online platform HRU-WEB. The HRU are used as modeling entities that have the same pedological, lithological, topographical, and land use/land cover characterizations, and are heterogeneous from each other. They are connected by a topological routing scheme [63]. The lateral water flow is simulated allowing a fully explicit spatial discretization of hydrologic response within the modeled catchment. However, the delineation of HRUs may not account for karst heterogeneity. Nevertheless, the model can be calibrated so that the observed and modeled stream flows match well.

The parameterization of input data was accomplished in step four and utilized computer programs, research articles and technical studies where lithologic, soil and land use properties were calculated or indicated. The selected values of all input parameters as well as the corresponding sources of information are listed in Table 2, Table 3 and Table 4. For every geologic unit, a distinction was made between the upper groundwater reservoir composed of loose weathered material with high permeability and the lower ground-water reservoir comprising the fractures and clefts of the bedrock. Consequently, two basis runoff components are generated: a fast one from the upper groundwater reservoir and a slow one from the lower groundwater reservoir. The filling of the groundwater reservoir results from the vertical runoff component (percolation) of the soil module. The parameterization of groundwater reservoirs is carried out with the definition of the maximum storage capacity of the upper and the lower groundwater reservoir as well as a retention coefficient each for both reservoirs and (Table 4).

The fifth step involved the execution of JAMS J2000 modules related to model initialization, estimation of rainfall interception, soil water, groundwater, and stream flow routing. It also comprised the calibration and validation of results, based on the NSIN II algorithm (Genetic Multi-objective II) with daily time step, while adopting 5000 iterations as stopping rule [70]. The specific modules are listed as Appendix A.

Following step five and the calibration/validation procedures, the hydrological model was tested for performance determined by comparison of observed versus simulated hydrographs as well as assessment of four goodness-of-fit indices: a) percentage of bias (PBIAS); b) coefficient of determination (R^2^); c) Nash–Sutcliffe (NSE) efficiency coefficient; and d) the natural logarithm of the NSE coefficient (LNSE).

According to Gupta et al. [71], the PBIAS estimates the percentage trend of simulated data to be higher or lower than the observed data and can be described by the following equation:
(1)$$PBIAS=\left[\frac{\sum_{t=1}^{n}(y_{i}-o_{i})}{\sum_{t=1}^{n}o_{i}}\right]\times 100$$
where PBIAS is the percentage of bias (%), yi is the simulated flow (m^3^/s), and oi is the observed flow (m^3^/s). A PBIAS = 0 occurs for a hydrological model with optimal performance. Positive or negative values indicate, respectively, that the model overestimates or underestimates the simulated flows.

R^2^ is a statistical test indicating the linear dispersion between the observed and simulated flows and can be expressed by the following equation [72]:
(2)$$R^{2}={\left(\frac{\sum_{i=1}^{n}\left(o_{i}-O^{med}\right)\left(y_{i}-Y^{med}\right)}{\sqrt{\sum_{i=1}^{n}{\left(o_{i}-O^{med}\right)}^{2}}.\sqrt{\sum_{i=1}^{n}{\left(y_{i}-Y^{med}\right)}^{2}}}\right)}^{2}$$
where R^2^ is the coefficient of determination (dimensionless) and o_i_, O_med_, y_i,_ and y_med_ represent, respectively, the observed flow, the average observed flow, the simulated flow, and the mean simulated flow, all expressed as m^3^/s. The interval of this test is between 0 and 1, where 0 means that there is no correlation between the simulated and observed values, and 1 indicates that the simulated values are the same as those observed. In this sense, the larger the value of R^2^ the greater the hydrological model efficacy.

The value of R^2^ is supposed to accommodate all sources of uncertainty and error usually involved in the hydrologic modeling of catchments, which are numerous and have been fully described elsewhere [73,74,75]. For example, estimating subsurface flow from soil porosity obtained from soil maps is a big source of uncertainty and error, while any hydrological model has a lot of error in the estimation of evapotranspiration from forests because of species-level differences in transpiration. It is expected that the model handles all errors at once by forcing simulated surface flows to equal observed surface flows. It is however worth noting that even measured streamflow time series that are commonly derived from stage-discharge rating curves, are themselves affected by the uncertainty of rating curves. While different methods to quantify uncertainty in the stage–discharge relationship exist, there is limited understanding of how uncertainty estimates differ between methods due to different assumptions and methodological choices [76].

The conjunctive use of R^2^ with other statistical analyses is recommended, since R^2^ only estimates the linear dispersion between observed and simulated data, disregarding minimum and maximum flow variations in the hydrological models [77]. The same authors suggested the use of the NSE coefficient or its natural logarithm (LNSE), since these coefficients describe the sum of absolute differences between observed and simulated flows during the studied period, representing the basin responses to base and peak flows. The NSE and LNSE coefficients are equated as follows [78]:
(3)$$NSE=1-\frac{\sum_{i=1}^{n}{\left(o_{i}-y_{i}\right)}^{2}}{\sum_{i=1}^{n}{\left(o_{i}-O^{med}\right)}^{2}}$$
(4)$$LNSE=1-\frac{\sum_{i=1}^{n}{\left(lno_{i}-lny_{i}\right)}^{2}}{\sum_{i=1}^{n}{\left(lno_{i}-lnO^{med}\right)}^{2}}$$
where “ln” represents the natural logarithm and oi, omed, yi, and ymed represent, respectively, the observed flow, the average observed flow, the simulated flow, and the mean simulated flow (m^3^/s). The values of NSE and LNSE (dimensionless) can vary from −∞ to 1. The closer to 1, the greater the adjustment between the simulated and observed values. Results below 0 indicate that the mean observed values are more representative than the values predicted by the model.

In this study, performance analysis was based on PBIAS and NSE coefficients. The reference values for performance levels are depicted in Table 5. The evaluation of performance was done separately for the calibration period (2003 to 2011), validation period (2012 to 2016), and full data period (2003 to 2016), allowing the verification of the replication of parameters in basins with similar characteristics as regards soil, land use, geology, relief and climate. The final outcome of the JAMS J2000 model was calibrated and spatially distributed flow components, namely the surface flow, the percolation in the soil, and the groundwater flow in the upper and lower parts of the aquifer.

## 3. Results

The modeled sub-basin of Jequitibá River basin is located upstream of hydrometric station 41410000. This sub-basin is represented in Figure 1 and corresponds to the upper part of the Jequitiba River basin. During the hydrologic modeling, the sub-basin was discretized in 22,920 hydrologic response units (HRU), which are the aforementioned modeling entities characterized by similar soil types, relief classes, and land uses/occupations. The HRU attributes, together with the climate data were submitted to 5000 iterations in JAMS J2000 modules, which returned values for specific hydrologic parameters as listed the Appendix B. Among the 24 input parameters (see the Appendix B), only three presented values were very close to the upper threshold, namely mFCa (4.99), the mace (4.98), and soilConcRD1 (9.99). All these parameters are related to the physical characteristics of soils. A higher value of maximum infiltration is reported between the months of April and September (soilMaxInf1 = 129.97 mm), than between the months of October and March (soilMaxInf2 = 75.99 mm), which can be explained by saturation conditions of meso- and macropores of soils in the rainy months of the year.

The relationship between observed and simulated flows in the 2003–2016 period is illustrated in Figure 3. The simulated hydrograph is very similar to the observed hydrograph. The simulated maximum, mean, and minimum flows were: 30.8 m^3^ s^−1^, 2.34 m^3^ s^−1^, and 0.19 m^3^ s^−1^, respectively. These results were close to the observed flows, which were 28.2 m^3^/s (maximum), 2.26 m^3^/s (average), and 0.24 m^3^/s (minimum). The results of model performance are displayed in Table 6. Whole period values of PBIAS indicate a very good performance while NSE values indicate a good/fair level. Overall, a good performance can be assumed.

The seasonal distribution of flow components is illustrated in Figure 4. On a monthly basis, surface flow (ED) presented an average value of 127.59 mm, and a very wide range (1.12–405.81 mm). These values represent water that has not been intercepted, has not evaporated, and has not infiltrated the soil. On the other hand, the subsurface flow (ES), which occurs between the soil layers, presented an average value of 33 mm and a maximum value of 149 mm, indicating a more constant flow than the first, although with smaller peaks. The upper underground flow (ESUBsup) presented a mean value of 18 mm and a maximum value of 39 mm, while the lower underground flow (ESUBinf) presented mean and maximum values of 7 mm and 17 mm, respectively.

The spatial distribution of flow components is displayed in Figure 5. It is clear from this figure that surface flows dominate in the northwestern sector of the modeled sub-catchment where the Sete Lagoas town is located. This is obviously explained by impervious surfaces in this urban area. Concomitantly, the other flow components are lower in this sector, especially the deep groundwater flow.

A cross tabulation of Figure 5 with the land use, soil, and geology maps allowed the quantification of flow components within the corresponding soil, land use, and geology types. The results are portrayed in Figure 6. The conclusion taken from the observation of Figure 5 is clearly demonstrated in Figure 6a where surface flows dominate in the urbanized areas and reach 228 mm every year, on average. Urbanization reduced substantially the shallow groundwater flow, which is 54 ± 5 mm for all land uses except the urban use where it is just 60% of that value (32 mm). The low topographic relief in the urban areas may also help to explain these values since there are less hilly areas in the city as compared to the hills in the southeast of the basin. Bare land has the largest flow in the soil, but this land use type is represented in just a small portion within the sub-basin headwaters (see Figure 1e). For the other more represented land use types, the annual percolation in the soil is larger in the areas used for agriculture (107 mm) and pasture (102 mm), represented as antropized areas in Figure 1e, and lower in the planted (67 mm) and native vegetated (Cerrado; 65 mm) areas. It should be noted, however, that the areas used for agriculture or pasture are usually located where relief is flat or smoothly undulating, favoring infiltration, while forests are frequently planted in sloping hillsides to protect soil from water erosion and native vegetation is widespread along the water courses, in areas where the steeper slopes do not favor infiltration and hence soil percolation. The most represented soil types (latosols and cambisols; see Figure 1d) distribute differently the flow components: every year, on average, the areas covered by latosols contribute strongly to surface flow (87 mm) and percolation in soil (88 mm), but little to shallow (25 mm) and deep (11 mm) groundwater flow, the cambisols contribute less to the first two components (64 mm for surface flow and 69 mm for percolation in the soil) but much more to the shallow groundwater flow (55 mm) and more to the deep groundwater flow (20 mm). Flow in the various lithologic units is similar, decreasing from surface flow (91.4 ± 15.9 mm), to percolation in soil (79.8 ± 15.7 mm), to shallow groundwater flow (46.3 ± 10.3 mm), and finally to deep groundwater flow (46.3 ± 10.3 mm). There are, however, some points to refer to. Percolation in soil is substantially larger in the limestones (102 mm) than in the other lithologic types (72.4 ± 6.0 mm). The other underground components are also larger in the limestones (55 mm for the shallow groundwater flow and 20 mm for the deep groundwater flow), although the differences to the other geologic units are smaller (43.4 ± 10.5 mm and 18.5 ± 0.9 mm, respectively).

## 4. Discussion

In 1993, the 150,000 inhabitants of Sete Lagoas consumed 200 L day^−1^ of karst aquifer water. This represented an annual abstraction of approximately 10.95 hm^3^ of groundwater [54]. Twenty-five years later (in 2008), the abstraction of groundwater resources increased to 14.6 hm^3^ [56]. Based on the hydrologic modeling, the annual groundwater resources within the Sete Lagoas karst (14.4% of the modeled catchment) represented in the 2003–2016 period, 6.3 hm^3^ yr^−1^. The difference between availability and demand is therefore negative and changed from −4.95 hm^3^ yr^−1^ in 1993 to −8.3 hm^3^ yr^−1^ 2008. The available karst groundwater resources in the entire studied area (42.7% of the modeled catchment) are 18.6 hm^3^. Based on these results, it can be suggested that the karst aquifer within the Sete Lagoas town has been overexploited for many years, because the renewable (through annual recharge) groundwater resources are much less than the annual abstractions. This substantial and expanding negative balance between renewable groundwater resource and abstraction is in keeping with the reports of hydraulic head declines occurring in Sete Lagoas town since 1942 [57]. At regional scale (considering the karst within the modeled sub-basin) the renewable resources are still enough to supply the demand (18.6 hm^3^ >14.6 hm^3^). As corollary of this analysis, it is also suggested that abstraction in the 94 drilled wells used for public water supply to Sete Lagoas is forcing groundwater flow in the direction of Sete Lagoas, from karst areas away from the town, eventually contributing to generalized water table declines in the karst aquifer. As documented for the Sete Lagoas town, hydraulic head declines have triggered the development of suffosional sinkholes. The expansion of hydraulically depressed areas caused by continued and excessive pumping can expand the zone of influence of these geo hazards. There are numerous studies relating overexploitation with water table declines and development of sinkholes [79,80,81,82,83]. According to these authors, excessive abstractions generate steep cones of depression that accelerate groundwater flow towards them, while slow phreatic recharge is replaced by more rapid downward percolation favoring suffosion, especially when the water table is lowered below the rock head. These two processes are accompanied by increases in the effective weight of the sediments due to loss of buoyant support that ultimately leads to sinkhole formation.

The modeling results (Figure 6a) associated a very large surface flow to the urbanized areas (228 mm yr^−1^), while the other land uses were related to relatively low runoff (on average, 44 ± 19 mm yr^−1^). The influence of urbanization on the surface water component can be interpreted as the difference between the two values, which means that urbanization has probably raised runoff by some 184 mm yr^−1^, or 7.9 hm^3^ yr^−1^ within the Sete Lagoas town. If this runoff could have been converted into recharge, the current impairment between groundwater resources and water demand would be less serious. Strom water infiltration is practiced in many urban areas to promote recharge and re-establish pre-urbanization hydrology [84,85], but this is barely recommended when the underlain aquifer is a karst. Urban storm water can transport significant amounts of sediments and contaminants (metals, hydrocarbons) that readily reach the karst aquifer because of its large permeability and preferential flow paths [7,86,87,88,89]. Besides groundwater contamination, storm water infiltration can also promote suffosional sinkhole formation. The influence of urbanization in sinkhole development was studied by White et al. in central Pennsylvania [90], who related suffosional loss of soil cover and consequent sinkhole formation with runoff concentration from paved roads and roofs in very localized areas, namely storm water retention dry wells.

The surface water component could however be used to complement the public water supply, if stored in small dams in the forested areas of the catchment headwaters [91]. The use of small dams minimizes negative consequences upon aquatic ecosystems, such as habitat fragmentation [92]. These areas represent 15.8% of the modeled sub-basin and runoff in these areas is approximately 50 mm yr^−1^. The surface water resource could therefore reach 1.9 hm^3^ yr^−1^. This conjunctive water management involving public water supply with good quality surface and groundwater resources is suggested for other regions [93,94,95,96,97,98] and could help to reduce the pumping in the Sete Lagoas drilled wells to mitigate subsidence and suffosional sinkhole formation, as proposed by Galvão [57].

The modeling results indicated a large soil percolation in the limestones (Figure 6c). The results also suggested a relationship between large percolation and coverage of limestones by permeable latosols (Figure 6b) used for agriculture and pasture (Figure 6a). It is important to preserve infiltration capacity in the limestone areas to keep the karst aquifer refilled with soil water. It is however worth noting that, in karst areas, increasing the water input to the ground increases percolation accelerating suffusion, favors dissolution, increases the weight of sediments, and may reduce the mechanical strength and bearing capacity of sediments. The isolated or combined effect of such processes can be the development of suffosional sinkholes with impacts on ground activities due to land disruption, namely agriculture. There are various studies relating the development of sinkholes with the irrigation of crop areas [99,100], which indicated the replacement of conventional furrow irrigation by sprinkler irrigation as a mitigation measure. Additional recommendation is about implementation of Better stated as Best Management Practices based on adequate decision support systems that minimize soil erosion and nutrient loss, as well as judiciously applying fertilizer to avoid water contamination [101,102,103].

Overall, the modeling results confirmed the diagnosis of overexploitation reported in earlier studies on the hydrology Jequitiba River basin and Sete Lagoas karst aquifer, while helping to find solutions for sustainable water management in this region.

## 5. Conclusions

The upper part of the Jequitiba River basin (state of Minas Gerais, Brazil) is shaped on crystalline and carbonate rocks and occupied by the Sete Lagoas town, a densely urbanized area with approximately 200,000 inhabitants. For decades, this town has been supplied with groundwater abstracted from a karst aquifer developed on the carbonate rocks. The results of a hydrologic modeling using the JAMS 2000 software revealed that the abstractions largely exceed renewable groundwater resources accomplished through recharge, the reason why the aquifer can be considered overexploited. The excessive abstractions have been recognized before, in studies focused on the spatial analysis of water table drawdowns. These earlier studies also reported the occurrence of geo hazards related to development or acceleration of suffosional sinkholes.

The hydrologic modeling allowed quantification of flow components, namely surface flow, percolation in soil and groundwater flow. The surface flow in the urbanized area is five times higher than in other land occupations. This result was expected, but the amount of surface water that is being hampered from infiltration (7.9 hm^3^ yr^−1^) is expressive. Since the use of storm water systems to artificially infiltrate this excess water is not recommended for environmental reasons, namely the high risk of groundwater contamination with metals and hydrocarbons transported in storm water, it was suggested to store quality surface water in forested areas from the catchment headwaters, using small dams. The estimated amount of surface water storable in these areas is 1.9 hm^3^ yr^−1^. This surface water could be diverted to and used in the urban area as a complement to the groundwater supply, in the context of conjunctive water resources management. Conjunctive water resources management would also enable the reduction of pumping rates and times at the drilled wells used for public water supply to Sete Lagoas town.

A relevant result from the hydrologic modeling was also the large contribution of the soil water component to aquifer recharge in the karst area, occurring in areas used for agriculture and pasturing. Fertilizing in these areas may be abundant and therefore the potential for groundwater contamination is presumably high, unless best management practices are implemented and monitored in these areas. Irrigation plays a role in this context. Sprinkler irrigation is recommended because it can not only attenuate the migration of nutrients from soil to soil water and groundwater, but also prevent the occurrence of suffosional sinkholes. In karst areas, these geo hazards are frequently related to concentrated water in the soil and can disrupt the land with negative consequences for economic activities, namely agriculture.

## Figures and Tables

**Figure 1 ijerph-16-02542-f001:**
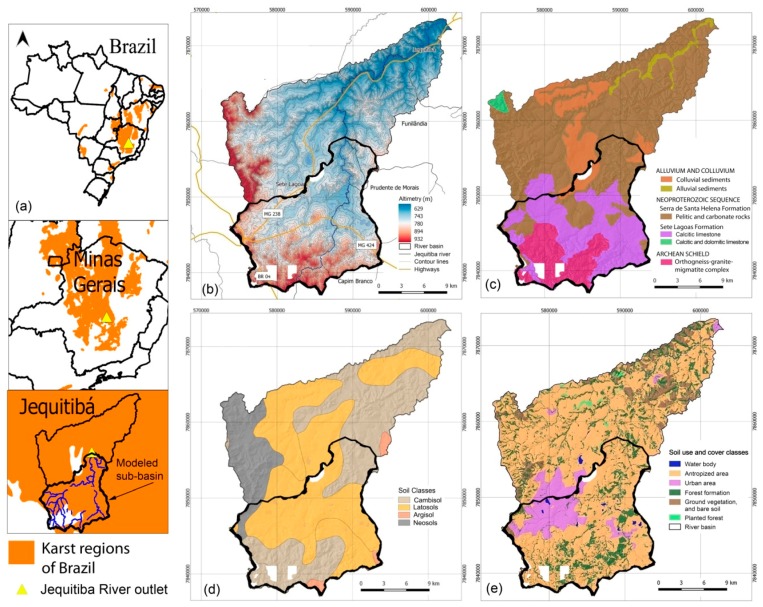
(**a**) Location of study area: Jequitiba River basin, Minas Gerais, Brazil; (**b**) topographic map of Jequitiba River basin, with indication of towns, drainage network, and main road network; (**c**) lithologic map of the Jequitiba River basin; (**d**) soil map of the Jequitiba River basin; (**e**) land use and cover map of the Jequitiba River basin; The geographic reference for the maps is the UTM projection system, SIRGAS 2000 datum, 23 south time zone.

**Figure 2 ijerph-16-02542-f002:**
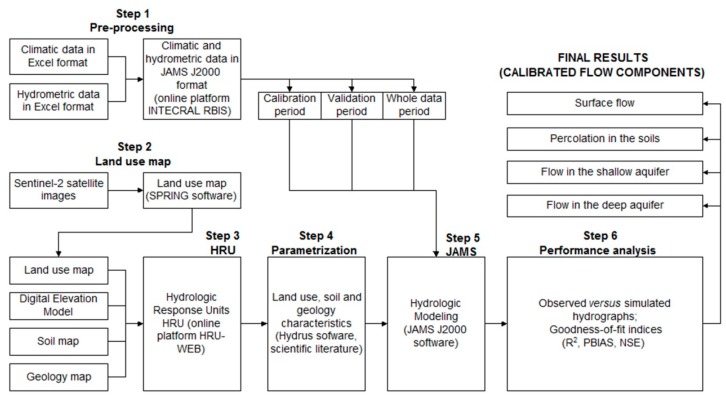
Workflow used to perform the hydrologic modeling.

**Figure 3 ijerph-16-02542-f003:**
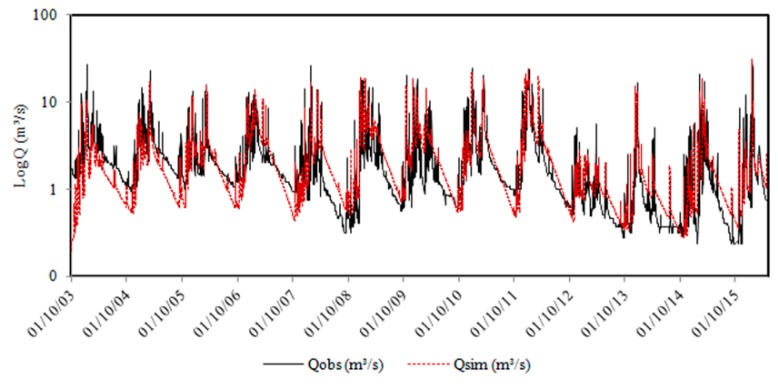
Hydrographs observed (Qobs) and simulated (Qsim) stream flows at the outlet of the Jequitibá River Basin.

**Figure 4 ijerph-16-02542-f004:**
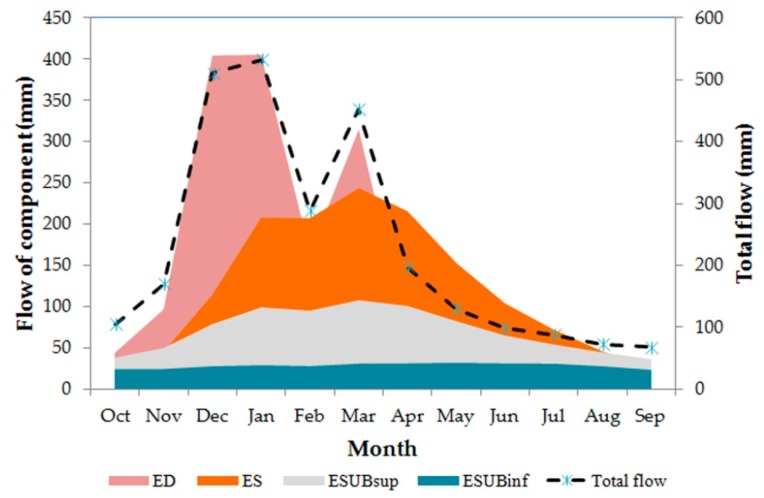
Seasonal distribution of total flow and of stream flow components estimated by the hydrological model. Symbols: ED—surface flow (overland flow due to sealing or saturation excess), ES—fast interflow (percolation within the upper soil layer); ESUBsup—fast baseflow (usually from the weathering part or fissures if existing); ESUPinf—base flow from base rock. Total flow = ED + ESUBsup + ESUBinf. The values represented in mm were calculated as amount of flow (m^3^) divided by HRU area (m^2^).

**Figure 5 ijerph-16-02542-f005:**
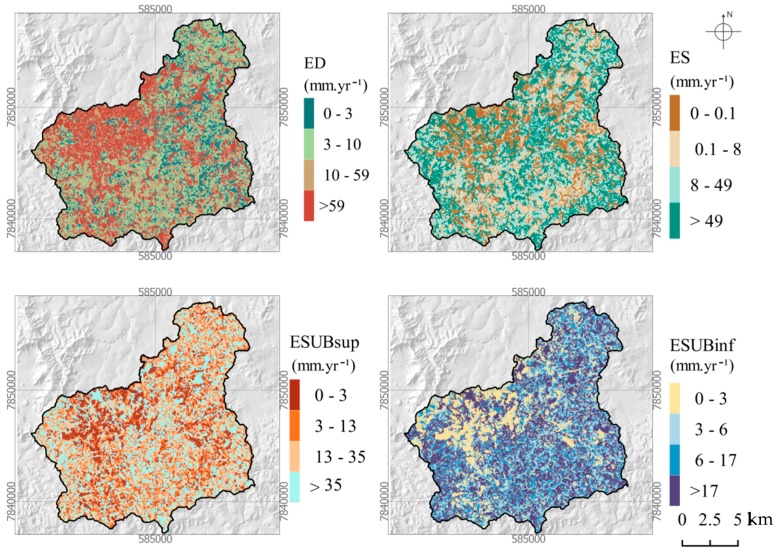
Spatial distribution of stream flow components estimated by the hydrological model. Symbols: ED—surface flow (overland flow due to sealing or saturation excess), ES—fast interflow (percolation within the upper soil layer); ESUBsup—fast baseflow (usually from the weathering part or fissures if existing); ESUPinf—base flow from base rock.

**Figure 6 ijerph-16-02542-f006:**
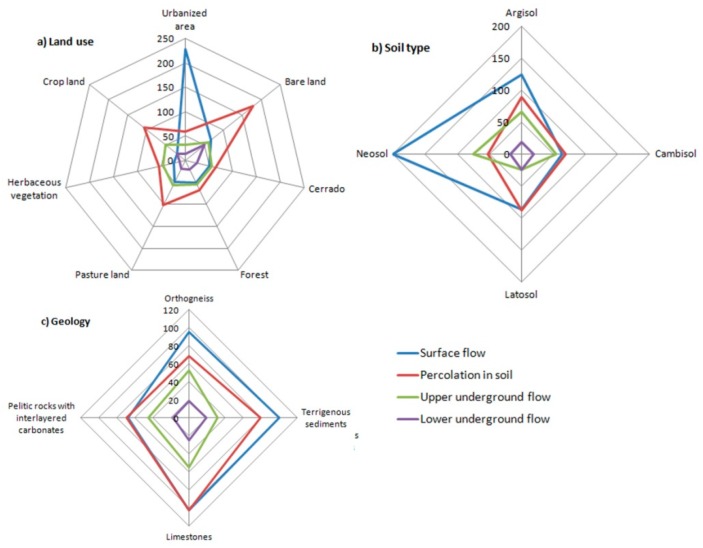
Distribution of stream flow components per land use (**a**), soil (**b**) and lithologic (**c**) types.

**Table 1 ijerph-16-02542-t001:** Materials used in the JAMS J2000 hydrologic model, namely spatial data and climatic and stream flow records, and URLs of websites used for downloading the data.

Data Type	Use in the Hydrologic Model	URL of Website
Digital elevation model	Hydrologic Response Units (HRU)	https://www.asf.alaska.edu
Satellite images	Land use mapping and HRU	https://earthexplorer.usgs.gov/
Soil map and hydraulic Conductivity data	HRU and data parameterization	http://www.dps.ufv.br
Geologic map	HRU and data parameterization	www.portaldageologia.com.br
Climatic data	Data for JAMS J2000 hydrologic model	http://www.inmet.gov.br
Stream flow data	Calibration/validation procedure	http://www.snirh.gov.br/hidroweb
Administrative data	Additional information	http://www.car.gov.br
Population data	Additional information	http://www.sidra.ibge.gov.br

**Table 2 ijerph-16-02542-t002:** Land use and occupation parameters used in the hydrologic model.

Land Use or Occupation	Albedo (%)	Superficial Resistance (s/m)	Leaf Area Index (Dimensionless)	Effective Growth (m)	Root Depth (cm)
Cultivated area	20.0	70.0	0.6	1.1	20.0
Urbanized area	16.4	70.0	0.01	0.0	0.0
Cerrado biome	14.2	70.0	0.8	20.0	120.0
Water bodies	4.0	70.0	0.0	0.0	0.0
Forest	15.0	70.0	0.9	30.0	300.0
Bare land	20.0	70.0	0.0	0.0	0.0
Reference(s)	[64,65]	[66]	[67]	[68]	[66]

**Table 3 ijerph-16-02542-t003:** Soil parameters used in the hydrologic model. The air capacity and field capacity (water holding capacity) are practical measures to describe the pore size differences between water that can be held against gravity (middle pore storage) and water that cannot (macro pore storage), respectively.

Soil Type	Depth (cm)	Minimum Permeability Coefficient (mm/d)	Air Capacity (mm)	Field Capacity (mm)
Red-yellow argisol	170	1	40	600
Haplic cambisols	230	1	37	1150
Red-yellow latossols	250	1	38	1500
Tholic Litholic	50	1	13	125
Reference	Hydrus 1D software (https://www.pc-progress.com)			

**Table 4 ijerph-16-02542-t004:** Lithologic parameters used in the hydrologic model.

Lithologic Type	Maximum Storage Capacity in the Upper Aquifer (mm)	Maximum Storage Capacity in the Lower Aquifer (mm)	Storage Coefficient in the Upper Groundwater Reservoir (d)	Storage Coefficient in the Lower Groundwater Reservoir (d)
Orthogneiss	50	900	13	365
Clastic sediments	50	800	16	365
Limestone	70	1000	17	365
Silstone	60	900	14	365
Reference	[69]			

**Table 5 ijerph-16-02542-t005:** Reference values of PBIAS and NSE and their relation with hydrologic model performance.

PBIAS (%)	NSE	Performance
0 a 10	0.75 a 1	Very good
10 a 15	0.65 a 0.75	Good
15 a 25	0.50 a 0.65	Fair
>25	<0.50	Inadequate

**Table 6 ijerph-16-02542-t006:** Results of hydrological model (JAMS J2000) performance analysis (values in %).

Performance Indicator	Evaluation Period	
Calibration	Validation	Whole Period	Performance(Whole Period)
PBIAS	−9.50	−3.65	3.80	Very good
R^2^	0.58	0.67	0.66	
NSE	0.58	0.67	0.64	Fair/Good
LNSE	0.62	0.60	0.60

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
