# Peer review of "Hydrologic Modeling for Sustainable Water Resources Management in Urbanized Karst Areas"

_ijerph, 2019, doi:10.3390/ijerph16142542_

Round 1

Reviewer 1 Report

I think this paper hold a very important topic on hydrological modelling in urban Karst areas.

The English and structure are very good for publication. Some minor comments should be address before accepting it.

Specific comments:

1.       Figure 2 is not clear for reader, can you make the font bigger?

2.       Figure 3, please change the green background into white.

3.       Figure 4, I suggest it should use line to present the flow.

4.     Please cite the following references:

Pingping Luo, Meimei Zhou, Hongzhang Deng, Jiqiang Lyu, Wenqiang Cao, Kaoru Takara, Daniel Nover, S. Geoffrey Schladow, Impact of forest maintenance on water shortages: Hydrologic modeling and effects of climate change, Science of the Total Environment, 615, pp. 1355-1363.

Pingping Luo, Dengrui Mu, Han Xue, Thanh Ngo-Duc, Kha Dang-Dinh, Kaoru Takara, Daniel Nover, Geoffrey Schladow,Flood inundation assessment for the Hanoi Central Area, Vietnam under historical and extreme rainfall conditions, Scientific Reports(Nature), 2018, 8:12623, DOI:10.1038/s41598-018-30024-5.

Pingping LUO, APIP, Bin He, Weili Duan, Kaoru Takara, and Daniel Nover: Impact assessment of rainfall scenarios and land-use change on hydrologic response using synthetic Area IDF curves, Journal of Flood Risk Management, Vol.11, pp.S84–S97, DOI: 10.1111/jfr3.12164, 2018.

Pingping LUO, Bin He, Kaoru Takara, Yin E Xiong, Daniel Nover, Weili Duan, and Kensuke Fukushi, Historical Assessment of Chinese and Japanese Flood Management Policies and Implications for Managing Future Floods, Environmental Science & Policy,Vol.48, 2015, pp. 265-277, DOI: 10.1016/j.envsci.2014.12.015.

Author Response

Response to Reviewer #1

I think this paper hold a very important topic on hydrological modelling in urban Karst areas.

The English and structure are very good for publication. Some minor comments should be address before accepting it.

We very much thank your nice appreciation. We will address all comments and suggestions you make below.

Specific comments:

1.       Figure 2 is not clear for reader, can you make the font bigger?

The original font size was 13px for all texts. Now is 15px for the texts located in the boxes and 16px for the boldface headings. That’s the maximum we could do.

2.       Figure 3, please change the green background into white.

The background was white and became green with the PDF conversion. Now the problem is solved.

3.       Figure 4, I suggest it should use line to present the flow.

The total flow was added to the figure, to the secondary YY axis, as suggested.

4.       Please cite the following references:

Pingping Luo, Meimei Zhou, Hongzhang Deng, Jiqiang Lyu, Wenqiang Cao, Kaoru Takara, Daniel Nover, S. Geoffrey Schladow, Impact of forest maintenance on water shortages: Hydrologic modeling and effects of climate change, Science of the Total Environment, 615, pp. 1355-1363.

Pingping Luo, Dengrui Mu, Han Xue, Thanh Ngo-Duc, Kha Dang-Dinh, Kaoru Takara, Daniel Nover, Geoffrey Schladow,Flood inundation assessment for the Hanoi Central Area, Vietnam under historical and extreme rainfall conditions, Scientific Reports(Nature), 2018, 8:12623, DOI:10.1038/s41598-018-30024-5.

Pingping LUO, APIP, Bin He, Weili Duan, Kaoru Takara, and Daniel Nover: Impact assessment of rainfall scenarios and land-use change on hydrologic response using synthetic Area IDF curves, Journal of Flood Risk Management, Vol.11, pp.S84–S97, DOI: 10.1111/jfr3.12164, 2018.

Pingping LUO, Bin He, Kaoru Takara, Yin E Xiong, Daniel Nover, Weili Duan, and Kensuke Fukushi, Historical Assessment of Chinese and Japanese Flood Management Policies and Implications for Managing Future Floods, Environmental Science & Policy,Vol.48, 2015, pp. 265-277, DOI: 10.1016/j.envsci.2014.12.015.

The references were all added to the revised manuscript.

Reviewer 2 Report

Dear Authors,

 this study is pertinent as an addition to the literature of hydrological modeling with direct application for water resources management in countries where such studies are not common. However, to be more useful to hydrologists and water resource practitioners, several improvements are necessary. In addition to the comments below, I have also entered comments on the attached pdf of your manuscript.

1.     In the introduction you mention that the heterogeneity of karst aquifers is a big challenge for estimating groundwater resources. Your study does not directly address that issue. You should make that clear, that the way the study estimates groundwater flows and stocks is by calibrating JAMS2000 to match observed surface flows over some years, then via validation over the next period of years, and bootstrapping to fine tune the model parameters to get a close match over the validation period. From your karstic heterogeneity inclusion, I was expecting to see some kind of new survey/tracer technique that maps karstic environments, but realized after reading the manuscript that this is an application of existing techniques. Which is fine, given that this is data for a part of the world that has not been modeled in this manner, and that this data can aid water resource management agencies.

2.     There is ambigiuity between percolation, shallow groundwater flow and deep groundwater flow. What are the bounds of shallow and deep aquifers? What seperates these – different hydraulic conductivity of aquifer geology? Is the shallow aquifer karst and deep aquifer the metamorphic rocks?

3.     Summing percolation, shallow flow and deep flow may be double-counting flow, because percolation adds to shallow aquifers and possibly deep aquifers if the two aquifers are connected.

4.     There needs to be sources and measures of uncertainty expressed for each of the terms. Observed streamflows have some uncertainty (in measurement and scaling up temporally and spatially) that will propagate into your model. Then estimating subsurface flow from soil porosity obtained from soil maps is another big source of uncertainty and error. You yourselves mention the heterogeneity and preferential flowpaths of karst, and this information does not appear to be present in you model inputs. Lastly, any hydrological model has a lot of error in estimation evapotranspiration from forests because of species-level differences in transpiration. Yes, a model accounts for that by forcing simulated surface flows to equal observed surface flows. However, readers will benefit from knowing what processes models cannot capture well, and the indirect approach taken by models to circumvent these deficiencies.

5.     Units – explain how you express flows in millimeters – is that on an annual time scale? Also be consistent in unit use, as sometimes you express in linear units (mm) and other times in flow ( m3/sec). Also distinguish between stocks and flows – it is not clear from the manuscript.

6.     How did you choose the duration of calibration and validation periods? Explain that in the methods. Note that the aim of a publication is to enable others to carry out similar exercises.

7.     The suggestion of building dams upstream of Sete Lagoas in forested headwaters can work only if the reservoirs are small, in order to avoid negative ecosystem consequences upon aquatic ecosystems. Another suggestion would be the use of constructed wetlands to store and treat a portion of urban runoff. Ideally, both approaches could be used.

8.     The English needs to be improved- while grammatically correct for the most part, the sentences are often long and the use of passive voice makes for awkward reading – one has to often read a sentence 2-3 times in order to comprehend the message. Also avoid words like “notorious”, “striking” as these are more relevant to literature than a scientific paper.

Having said all this, you can see that these are minor improvements. I hope these suggestions are useful.

Author Response

Response to Reviewer #2

this study is pertinent as an addition to the literature of hydrological modeling with direct application for water resources management in countries where such studies are not common. However, to be more useful to hydrologists and water resource practitioners, several improvements are necessary.

We very much thank the effort made by the reviewer and the positive appreciation regarding the pertinence of our study. We addressed all comments and suggestions as explained below.

In addition to the comments below, I have also entered comments on the attached pdf of your manuscript.

We replied to the comments in the PDF file where they were made and adjusted the revised text accordingly.

1.       In the introduction you mention that the heterogeneity of karst aquifers is a big challenge for estimating groundwater resources. Your study does not directly address that issue. You should make that clear, that the way the study estimates groundwater flows and stocks is by calibrating JAMS2000 to match observed surface flows over some years, then via validation over the next period of years, and bootstrapping to fine tune the model parameters to get a close match over the validation period. From your karstic heterogeneity inclusion, I was expecting to see some kind of new survey/tracer technique that maps karstic environments, but realized after reading the manuscript that this is an application of existing techniques. Which is fine, given that this is data for a part of the world that has not been modeled in this manner, and that this data can aid water resource management agencies.

The reviewer is right and we thank the comment. We rephrased the text moving the focus from karst heterogeneity, which was not specifically addressed in this study, to hydrologic modeling as way to evaluate groundwater resources in catchments that are frequently heterogeneous as regards many water balance parameters.

2.       There is ambigiuity between percolation, shallow groundwater flow and deep groundwater flow. What are the bounds of shallow and deep aquifers? What seperates these – different hydraulic conductivity of aquifer geology? Is the shallow aquifer karst and deep aquifer the metamorphic rocks?

We very much appreciate this pertinent comment. We clarified the ambiguity in the revised text. The following sentence was added:

For every geologic unit, a distinction was made between the upper groundwater reservoir composed of loose weathered material with high permeability and the lower ground-water reservoir comprising the fractures and clefts of the bedrock. Consequently, two basis runoff components are generated: a fast one from the upper groundwater reservoir and a slow one from the lower groundwater reservoir. The filling of the groundwater reservoir results from the vertical runoff component (percolation) of the soil module. The parameterization of groundwater reservoirs is carried out with the definition of the maximum storage capacity of the upper and the lower groundwater reservoir as well as a retention coefficient each for both reservoirs and (Table 4).”

3.       Summing percolation, shallow flow and deep flow may be double-counting flow, because percolation adds to shallow aquifers and possibly deep aquifers if the two aquifers are connected.

The reviewer is right but we took that into account. To clarify that, we modified Figure 4 and represented Total flow = ED + ESUBsup + ESUBinf, besides the representation of percolation.

4.       There needs to be sources and measures of uncertainty expressed for each of the terms. Observed streamflows have some uncertainty (in measurement and scaling up temporally and spatially) that will propagate into your model. Then estimating subsurface flow from soil porosity obtained from soil maps is another big source of uncertainty and error. You yourselves mention the heterogeneity and preferential flowpaths of karst, and this information does not appear to be present in you model inputs. Lastly, any hydrological model has a lot of error in estimation evapotranspiration from forests because of species-level differences in transpiration. Yes, a model accounts for that by forcing simulated surface flows to equal observed surface flows. However, readers will benefit from knowing what processes models cannot capture well, and the indirect approach taken by models to circumvent these deficiencies.

Many thanks for the pertinent comment. The following sentence was added to the revised ms, close to the presentation of performance indicators (R2).

The value of R2 is supposed to accommodate all sources of uncertainty and error usually involved in the hydrologic modeling of catchments, which are numerous and have been fully described elsewhere [73–75]. For example, estimating subsurface flow from soil porosity obtained from soil maps is a big source of uncertainty and error, while any hydrological model has a lot of error in the estimation of evapotranspiration from forests because of species-level differences in transpiration. It is expected that the model handles all errors at once by forcing simulated surface flows to equal observed surface flows. It is however worth to note that even measured streamflow time series that are commonly derived from stage‐discharge rating curves, are themselves affected by the uncertainty of rating curves. While different methods to quantify uncertainty in the stage‐discharge relationship exist, there is limited understanding of how uncertainty estimates differ between methods due to different assumptions and methodological choices [76].”

5.       Units – explain how you express flows in millimeters – is that on an annual time scale? Also be consistent in unit use, as sometimes you express in linear units (mm) and other times in flow ( m3/sec). Also distinguish between stocks and flows – it is not clear from the manuscript.

Except in Figure 3 where we compared measured with simulated stream flows in m3/s, the modeling results are all represented in mm, which is the annual discharge volume (m3) divided by the reference area (e.g. HRU; m2). This is clarified in the revised text.

6.       How did you choose the duration of calibration and validation periods? Explain that in the methods. Note that the aim of a publication is to enable others to carry out similar exercises.

We added the following sentence to the methods section to clarify our option:

The whole data period (14 years) was defined on the basis of available data. Within this period, a larger time span (9 years) was ascribed to the calibration period to allow improved hydrologic parameters, while the time window for validation was restricted to the remaining 5 years

7.       The suggestion of building dams upstream of Sete Lagoas in forested headwaters can work only if the reservoirs are small, in order to avoid negative ecosystem consequences upon aquatic ecosystems. Another suggestion would be the use of constructed wetlands to store and treat a portion of urban runoff. Ideally, both approaches could be used.

Yes. We are aware of that. We added some references to reinforce that conjunctive water resources management the way we hypothesize is based on small dams.

8.       The English needs to be improved- while grammatically correct for the most part, the sentences are often long and the use of passive voice makes for awkward reading – one has to often read a sentence 2-3 times in order to comprehend the message. Also avoid words like “notorious”, “striking” as these are more relevant to literature than a scientific paper.

We improved the English throughout the entire manuscript.

Having said all this, you can see that these are minor improvements. I hope these suggestions are useful.

The comments and suggestions, both the ones presented above and those inserted in the PDF file were all welcome and we did our utmost to improve the manuscript according to them.
